# Development of a Longitudinal Diagnosis and Prognosis in Patients with Chronic Kidney Disease: Intelligent Clinical Decision-Making Scheme

**DOI:** 10.3390/ijerph182312807

**Published:** 2021-12-04

**Authors:** Chin-Chuan Shih, Ssu-Han Chen, Gin-Den Chen, Chi-Chang Chang, Yu-Lin Shih

**Affiliations:** 1Dean of the Lian-An Clinic, Taipei 24200, Taiwan; joannayang@usmg.com.tw; 2Deputy Chairman, Taiwan Association of Family Medicine, Taipei 24200, Taiwan; 3Department of Industrial Engineering and Management, Ming Chi University of Technology, New Taipei City 243303, Taiwan; ssuhanchen@mail.mcut.edu.tw; 4Center for Artificial Intelligence & Data Science, Ming Chi University of Technology, New Taipei City 243303, Taiwan; 5Institute of Medicine, Chung Shan Medical University, Taichung 40201, Taiwan; gdchentw@hotmail.com; 6Department of Medical Informatics, Chung Shan Medical University & IT Office, Chung Shan Medical University Hospital, Taichung 40201, Taiwan; 7Department of Information Management, Ming Chuan University, Taoyuan 33300, Taiwan; 8Department of Otolaryngology-Head and Neck Surgery, Chang-Gung Memorial Hospital, Linkou Branch, Taoyuan City 33305, Taiwan; 20130030ryan@gmail.com

**Keywords:** chronic kidney disease, machine learning, risk prediction, clinical decision-making

## Abstract

Previous studies on CKD patients have mostly been retrospective, cross-sectional studies. Few studies have assessed the longitudinal assessment of patients over an extended period. In consideration of the heterogeneity of CKD progression. It’s critical to develop a longitudinal diagnosis and prognosis for CKD patients. We proposed an auto Machine Learning (ML) scheme in this study. It consists of four main parts: classification pipeline, cross-validation (CV), Taguchi method and improve strategies. This study includes datasets from 50,174 patients, data were collected from 32 chain clinics and three special physical examination centers, between 2015 and 2019. The proposed auto-ML scheme can auto-select the level of each strategy to associate with a classifier which finally shows an acceptable testing accuracy of 86.17%, balanced accuracy of 84.08%, sensitivity of 90.90% and specificity of 77.26%, precision of 88.27%, and F1 score of 89.57%. In addition, the experimental results showed that age, creatinine, high blood pressure, smoking are important risk factors, and has been proven in previous studies. Our auto-ML scheme light on the possibility of evaluation for the effectiveness of one or a combination of those risk factors. This methodology may provide essential information and longitudinal change for personalized treatment in the future.

## 1. Introduction

The progression of chronic kidney disease (CKD) is multifactorial and complex, proper management of CKD to slow the progression of this condition is of considerable significance. According to the Global Burden of Disease (GBD) study 2017, CKD resulted in 1.2 million deaths and was the 12th leading cause of death worldwide [1]. Based on the Taiwanese Ministry of Health and Welfare’s annual report, CKD accounts for the largest number of health insurance claims in 2018 [2]. In the 2019 annual report of the US Renal Registry System (USRDS) [3], Taiwan has the highest prevalence and incidence of end-stage renal disease in the world [4].

In consideration of patterns of CKD progression, it is critical to conduct risk diagnosis and prognosis for CKD patients. Moreover, CKD risk factors, such as hypertension, age, eGFR, UPCR, Smoking, obesity [5,6,7,8]. Addressing longitudinal risk factors for the progression of CKD is needed to reduce its associated morbidity and mortality. It’s not easy to detect chronic renal failure before losing 25% of renal function. Early prediction can possibly prevention, or dampen the progression of CKD to end-stage. According to the 2017 medical expenses of National Health Insurance Administration [4], the national health insurance expenditure with end-stage kidney disease increased year by year. The national health insurance expenditure increased from NTD. 295 billion in 2000 to NTD. 573.93 billion in 2016, with an average of nearly NTD. 500,000 in health insurance per year for each dialysis patient.

The most measure of kidney function, the eGFR, plays a critical role in CKD progression [2]. However, no obvious symptoms were found in an early stage of kidney disease. Thus, the clinical condition is usually asymptomatic until in advanced stages. Evidence on the convincing evidence of CKD screening is inadequate. Remembering “eGFR” as an estimate and not the measured GFR is important. Risk factors of CKD diagnosis and prognosis were extensively examined in recent years, but they are still controversial [9,10,11,12]. Many epidemiological studies showed a close relationship between hypertension and renal diseases. Early studies believed that effective hyperlipidemia treatment reduced proteinuria in patients with CKD, thus delaying renal function deterioration. However, research evidence has yet to prove the clear effect of hyperlipidemia on renal diseases. Hypercholesterolemia and hypertriglyceridemia are common in patients with nephrotic syndrome. Significantly elevated apolipoprotein B’s lipoprotein level, including very-low-density lipoprotein, intermediate-density lipoproteins, and low-density lipoproteins, as well as normal or slightly lower high-density lipoprotein levels, are usually detected in the blood of patients with nephrotic syndrome [13]. Recent studies found that catabolism reduction, decomposition, and lipoprotein removal not only play an important role but are partly associated with lipoprotein synthesis promotion. Some recent studies pointed out that severity reduction of proteinuria also reduces renal failure. Patients with renal insufficiency or severe proteinuria should be given with angiotensin-converting enzyme inhibitor or angiotensin receptor blocker [14].

Until now, few studies assessed the longitudinal assessment of multiple comorbidities of patients over an extended period, considering the CKD progression heterogeneity. Conducting a longitudinal diagnosis and prognosis in patients with CKD is crucial. Thus, an auto-Machine Learning (ML) scheme was proposed in this study, including classification pipeline, cross-validation (CV), Taguchi method, and improved strategies to predict early CKD. Especially, this auto-ML scheme illuminates the possibility of effectiveness evaluation of one or a combination of those risk factors.

## 2. Materials and Methods

In this study, the basic components are summarized in Figure 1, consisting of four main parts: classification pipeline, CV, Taguchi method, and improved strategies. The association of classification pipeline and CV is described in Section 2.1, and nine strategies of model performance improvement are separately discussed in Section 2.2. Finally, integration of all components using the Taguchi method is introduced in Section 2.3.

### 2.1. The Classification Pipeline with CV

The discrimination of a patient with CKD progression into the third stage or not is a typical classification task. The classification and regression tree (CART) is chosen as the classifier due to the flooded categorical and ordered variables in our dataset, unnecessary prior data distribution assumption, and the ability of the tree-based method to deal with missing data and perform a little bit well on imbalanced datasets compared to other methods.

The basic flowchart of the CV classification task is shown in Figure 2. The original dataset comes from different sources, such as basic information, basic examination data, blood test, and daily medication preference of patients, which are finally wrangled in a tidy form where columns mean different features, as well as the class labels, while rows mean cases. Then the original dataset is divided into training and testing datasets with a specific separation rate, typically 6:4, 7:3, or 8:2. During the training period, the training dataset is preprocessed resulting in either increasing or decreasing the number of columns or extracting embedding or group features. Search for hyper-parameters of a classifier during the training process is the next concern. The different base classifier has a different number of hyper-parameters, thus manual selection of a good hyper-parameter combination is very difficult. The process of combing the skills of deciding a hyper-parameter search strategy, conducting the k-fold CV, and selecting an evaluation metric is the most commonly used method to tackle the problem in the practice.

The training dataset is randomly divided into k equal-sized folds. Of the k folds, the k − 1 folds are the real training dataset, whereas the remaining single fold takes the validation dataset role in turn. For each set of hyper-parameters, which was generated by a search strategy, classifier weights via the training dataset were repeatedly learned and a metric validation dataset for k times, which are finally averaged to produce an average metric value, were evaluated. The higher the average value of a metric is, the better the hyper-parameters will be. Feature importance is listed with the best model in mind. The tidy format of the testing dataset is the same as that of the training dataset. The features and labels of a testing dataset are separated in advance. The dataset of features is fed into the best model to get responses and compare with the answer correspondences to yield a testing confusion matrix. The testing balanced accuracy is finally calculated for model evaluation, which is the average of sensitivity and specificity from the confusion matrix.

### 2.2. Using Different Strategies to Improve the Training Balanced Accuracy

Different strategies are used to improve data analysis as shown in Figure 2. After careful consideration, nine strategies were summarized as shown below.

Strategy 1: missing values imputation. Missing values is a common problem in practice. Sometimes without a great modeling impact, but sometimes causing modeling difficulties or failure, even with the mechanism of the tree-based model to combat the problem to some extent. Therefore, should these missing values be filled or be ignored before modeling becomes a strategic option. This study used the bootstrap aggregation imputation [15], which fits a bagged tree alternately based on regression dependencies [16].

Strategy 2: the inclusion of the cross-product term of original features. The effect of a certain feature on dependent variables, affected by other features, suggests an interaction between them. All paired cross-product terms between features are applied with this strategy application [17,18].

Cross-product terms of features have more predictive power than the original ones, which potentially increase the model nonlinearity and grasp the interaction relationship between features. A large number of cross-product terms lead to an overfitting model; however, the interference is alleviated by conducting a feature selection algorithm. Employing cross-product terms as additional CKD deterioration features is unobvious to clinical diagnosticians, but contributes to finding a powerful model with unobvious terms that serve as novel deterioration status features of a lesion [19,20,21].

Strategy 3: the clustering feature addition. The clustering technique groups similar training cases and assigns new columns for clustering labels in the form of dummy variables to the original training dataset. During the testing period, a clustering label is allocated to each testing case by finding the minimum distance between the case and cluster centers. Clustering before classification is beneficial [22,23]. In this study, the k-means algorithm is used for the clustering training dataset and the corresponding optimal number of clusters is determined by the rank aggregation algorithm [24].

Strategy 4: the prominent feature selection. The course of dimensionality is the most challenging problem. Maintaining less but significant features increase the convergence speed and improve prediction quality. This study introduces the least absolute shrinkage and selection operator (Lasso) to select features with stronger explanatory power from existing features and remove features with multicollinearity [25,26].

Strategy 5: the original feature transformation. A transformation technique converts the original feature space into other lower-dimensional spaces. The new feature space regarded the combination of original features in each dimension as a base. Instead of using principal component analysis, this study adopts correspondence analysis (CA) to extract significant base vector sets from our categorical or ordered dataset, which better express the variability of original features. The trained feature reduction procedure was empirically proven useful as a classifier [27].

Strategy 6: the resampling of cases in the minority class. The class imbalance problem often occurs in clinical datasets that comprise a higher number of normal cases relative to a number of patients. The classifiers need to identify rare but important cases; however, they are biased toward the majority class and struggle for yielding a fair accuracy [28,29,30]. In this study, the prediction was improved through a resampling by oversampling technique application. The oversampling technique tries to balance the number of cases in each class throughout minority class cases replication.

Strategy 7: the boosting capability enhancement for the classifier. Boosting is a type of ensemble learning for primarily converting weak learners to strong ones [31]. In this study, the boosting classifier was considered using eXtreme Gradient Boosting (XGBoost) because of its effectiveness as a tree-based ensemble learning algorithm [32]. XGBoost is a flexible classifier, which provides lots of fine-tuned hyper-parameters, such that made better predictions. In recent years, many Kaggle champion teams used XGBoost to win the titles, which is also successfully used for various medical issues [33,34].

Strategy 8: searching hyper-parameters randomization. Grid search is a typical technique to search better hyper-parameters using a CV procedure for a given classifier. The term grid originates from the combination of all possible trial values in a grid manner. An interesting alternative is a random search, which implements uniform randomness over the hyper-parameters. The performance of random search in cases of several algorithms on different datasets [35].

Strategy 9: the comprehensiveness of evaluation metrics. The evaluation metric used in k-fold CV affects the hyper-parameter selection results. The accuracy is the most commonly used metric that measures the number of correctly classified cases, both positive and negative. However, the accuracy says nothing about the classification performance for each class and it works with a fixed classification threshold on the class probability. An interesting alternative is an area under the curve (AUC) in which the curve is the receiver operating characteristic. The AUC evaluates the overall performance of a classifier that simultaneously takes the performance of each class and a series of classification thresholds into consideration.

### 2.3. Choosing the Strategy Combination Automatically

Multiple strategies above are used in the training process to improve predictive model performance. However, no specified strategy combination is proven as the best, it depends on the available dataset. Thus a sensitivity analysis needs to be conducted while users are training a model. In this study, a known Taguchi method was established for choosing a recommended strategy combination, in which the strategies are regarded as the factors and each strategy only has two levels of use or not, whereas the CV balanced accuracy from the training dataset is used to measure each treatment. The Taguchi method rather than the traditional 2^k^ design of experiment (DOE) is used because the number of treatments required for 2^k^ DOE surges with the number of factors k, for example, 29 non-repeated treatments in our study will have 512 performed trials. In this light, such full factor treatments consider all interactions and need too many experiments, causing waste of computation and time. The Taguchi method uses the orthogonal arrays (OA) to reduce the number of treatments that are originally required while avoiding a decreasing experiment power that comes with adopting fractional factorial designs [36,37].

Considering the above-mentioned conditions, the effects of different types of strategies on the training balanced accuracy of our CKD training data are studied. The stages for executing a Taguchi method for nine factors at two levels are shown in Figure 3 and are described as the following:

Step 1: OA design matrix preparation. The Taguchi method with factors number and levels number designed based on this study will obtain an OA design matrix. In this design matrix, each column stands for strategies, each row stands for each treatment, number 1 in the matrix means the corresponding strategy is used, and number 0 means not used. In addition, another column in the design matrix record the training balanced accuracy obtained for each strategy combination.

Step 2: Training model evaluation of each treatment. With the given training dataset and strategy combination for better training balanced accuracy, the classifier endeavors to optimize its hyper-parameters to obtain the optimal training balanced accuracy. Repeating the above experiment on different strategy combinations with random order thoroughly collects experimental data of the Taguchi method.

Step 3: Recommended strategy combination selection. This study uses a larger-the-better signal-to-noise ratio (S/N) to maximize the training balanced accuracy. The S/N of each treatment was calculated based on Equation (1), average the S/N of each level for each factor, and then output the main effects plots. The final decision for strategy combination selection is made by observing the positive or negative slope of the main effects plot of each factor. Only the strategies with a positive slope of the main effects plot are adopted.
(1)$$S/N=-10\cdot \log\left(\sum \left(1/training\;balanced\;accuracy^{2}\right)/N\right)$$

Step 4: Evaluate and test prediction accuracy. With given testing data, mostly recommended strategy combination and optimal hyper-parameters, the classifier performs class prediction for unseen testing cases. Finally, the predictive accuracy is evaluated through testing balanced accuracy, additionally, information about feature importance is provided as an identification maneuver of CKD risk factors.

## 3. Results

In this section, our data were first manipulated into the tidy form and a series of data analysis procedures was conducted using a self-programming toolkit under the R environment with the main package of “caret,” “optCluster,” “quality tools,” “MLmetrics,” and “MLeval,” as well as their dependencies.

Data were collected from individual CKD case administration and care systems of 32 chain clinics and three special physical examination centers. The data collecting period is from 1 January 2015, to 31 December 2019, total 50,174 effective records. Referring to the CKD third-stage progression rate of 34.69%, the total number of the class of third-stage CKD progression that is less than the total number of another class of non-progression is easily observed, thus a class imbalance problem arises. Classifiers that are commonly used always have a bias toward the majority class.

Basic information: admission date, sex, and date of birth.

Examination data: date of examination, height, weight, systolic pressure, diastolic pressure, urine polymerase chain reaction (mg/gm), urine albumin-to-creatinine ratio (mg/gm), uric acid (mg/dL), serum creatinine (mg/dL), eGFR (Modification of Diet in Renal Disease), cholesterol (mg/dL), low-density lipoproteins (mg/dL), HbA1C (%), sugar AC (mg/dL), hemoglobin A1c, CKD stage, comorbidity, and smoking.

As observed in Section 2.2, many techniques provided by researchers improved the prediction. However, most of those researches select appropriate strategies by trial-and-error methods, thus a systemic procedure is rarely seen. This study has nine possible strategies, without idea whether each strategy needs to be adopted to associate with the model in this dataset. Thus, a sensitivity analysis is conducted throughout the Taguchi method.

During the training stage, 30,106 cases are used to train the model in which the rate of third-stage CKD deterioration in training data is approximately 34.69%. The Taguchi OA L_12_(2^9^) design matrix is selected to evaluate the effect of multiple strategies in training balanced accuracy. In the first to ninth columns of Table 1, ones or zeros represented the use or un-use of the corresponding strategy, respectively, whereas the last column in Table 1 represents the values of training balanced accuracy for each treatment. Possible savings are apparent, the same number of factors and levels examined with DOE required 512 treatments, whereas only 16 in the Taguchi method. The value of training accuracy for each treatment is also recorded in the last column of Table 1. Fluctuating training balanced accuracy is found among the treatments results from whether each strategy will be adopted or not.

The regression equation of the fitted model is described in Equation (2). A positive or negative effect on the task of maximizing the training balanced accuracy as a coefficient of factor is also positive or negative, respectively. The R squared score is at a good level of 84.16%, which means that we are above 84% from the proportion of variance explained by the fitted regression model.
(2)$$\begin{array}{cc} S/N=-2.8383 & +0.0791\times \mathrm{Clustering}+0.0771\times \mathrm{CrossTerm}+0.6546\times \mathrm{FeatureReduction}\\ & -0.8004\times \mathrm{FeatureSelection}+1.3982\times \mathrm{Imputation}-0.0429\times \mathrm{Ensemble}\\ & +0.1033\times \mathrm{EvaluationMetric}-0.7432\times \mathrm{Randomized}-0.1957\times \mathrm{Resampling} \end{array}$$

Recommended level of each factor was finally determined based on the nine main effect plots as shown in Figure 4. The main effect plots show how each strategy affects the S/N ratio of training balanced accuracy. A pink line connects the points across all strategy levels. The slopes of those pink lines indicate the relative magnitude of the strategy effects. As shown in Figure 4, the imputation strategy has the largest effect on the S/N ratio, followed by the feature reduction strategy, and followed by the randomization strategy. In addition, the training balanced accuracy is maximized when the strategies of clustering, cross term, feature selection, ensemble, AUC, and randomization are at their highest setting and those of feature reduction, imputation, and resampling are at their lowest setting. Based on this analysis of the Taguchi method, the manual selection of strategy combinations for improving the accuracy was alleviated.

## 4. Discussion

Based on the optimal model selected throughout the training process described above, approximately 20,068 cases were further fed for testing the model’s performance of the proposed method. The rate of third-stage CKD progression in testing data is also approximately 34.69%. In Table 2, the proposed auto-ML scheme auto-selects the level of each strategy to associate with a classifier, which finally shows an acceptable testing accuracy of 86.17%, balanced accuracy of 84.08%, a sensitivity of 90.90%, and specificity of 77.26%, precision of 88.27%, and F1 score of 89.57%. Further, comparing the performance of two naive situations, i.e., only CART or XGBoost classifier is used and none strategy is adopted, the CART yields a lower testing accuracy of 84.14%, balanced accuracy of 82.02%, sensitivity of 88.97%, and specificity of 75.06%, precision of 87.04%, and F1 score of 87.99%, whereas the XGBoost also yields a lower testing accuracy of 83.82%, balanced accuracy of 79.39%, sensitivity of 93.86%, and specificity of 64.92%, precision of 83.44%, and F1 score of 88.34%. From this model comparison experiment, it can be seen that the classification accu-racy of CART has reached the level of about 84%. Compared with CART, the classification accuracy of XGBoost has decreased, and the level of specificity has also been sacrificed. In this study, XGBoost was selected as the basic classifier, and with the help of other strate-gies, it can further improve the classification accuracy rate by about 2% and the predic-tions will not be biased towards the majority class. A graphical comparison via a receiver operating characteristic (ROC) curve is also shown in Figure 5, confirming that the proposed method provides an easy way to auto-find out a suitable model for a given dataset.

In addition, the variable importance is also assessed in Table 3 that is to show which features are more influential on rate of CKD patients with progression to third stage.

Creatinine is made from creatine, which comes from the diet and biosynthesis of the human body [38]. The kidney and the liver are the major organs involved in the biosynthesis of creatine in the human body [39]. In the kidney, the L-arginine: glycin amidinotransferase transfer the amino group of arginine to glycine to yield ornithine and guanidinoacetate acid (GAA) [40], which will be transported to the liver by circulation. The S-adenosyl-1-methionine: N-guanidinoacetate methyltransferase in the liver methylated the amidino group of GAA to produce creatine [41]. Finally, the creatine form biosynthesis and diet are brought to the muscle and catalyzed into creatinine [42,43], which will be excreted by the kidney via urine [5].

In the normal condition, the creatinine is produced at a steady rate. The kidney is the major organ excreting the creatinine. Creatinine is not reabsorbed and the tubular secretion of creatinine is negligible, thus the eGFR is calculated from the excretion of creatinine and represents the GFR. CKD is a renal disease with declined renal function especially filtration in the kidney. Therefore, the creatinine accumulates in the body of a patient with CKD, thus a higher creatinine level. The staging of CKD depends on the level of serum creatinine, so the patient with CKD must have a high serum creatinine [44]. This result is also corresponding to our study in Table 3. The creatinine level is the most influential factor among all the other factors in third-stage CKD. The creatinine level alone is the second most important risk factor, and creatinine level with other risk factors is an important risk factor in our study.

Age is another risk factor for CKD. After the age of 30 years, the glomerulus is replaced by fibrous tissue, and this process is called glomerulosclerosis. The mesangium increases to approximately 12% at the age of 70 years [45]. Meanwhile, the vessel formed between afferent and efferent arterioles causes a shunt, especially at the juxtamedullary nephrons. The other arterioles of the kidney thicken and lost autonomic vascular reflex. Renal tubules have fatty degeneration and thicken their basal membrane. As a consequence, the renal tubule and glomerulus become atrophy and fibrosis [7]. These factors impair the renal function of the elderly. In Table 3, age plays an important role in third-stage CKD. Age with creatinine level becomes the most influential risk factor among the others. Age alone and age with smoking also account as the eighth and tenth most influential risk factors in our study.

Hypertension is another risk factor. Glomerular hypertension causes endothelial damage and glomerular vascular stretching. Eventually, cause elevated leakage protein from the glomerulus, glomerular collapse, glomerulonecrosis, and necrosis [8] Renin-angiotensin-aldosterone system (RAAS) in hypertension also sabotage renal function. According to previous studies, angiotensin II along with other RAAS components triggers inflammation and fibrosis [46,47]. The damage of the arteriole, glomerulus, renal tubule and kidney tissue ultimately increases inflammation and oxidative stress. The final results are arteriosclerosis, glomerular injury, and tubule-interstitial fibrosis. All in all, hypertension exacerbates renal function, congruent in our study. In Table 3, hypertension with creatinine level, elevated diastolic blood pressure with creatinine level, and elevated systolic blood pressure with creatinine level was third, sixth, and ninth most influential risk feature in third-stage CKD, respectively.

Smoking is notorious for vascular injury and damages renal function. Smoking can compromise the renal function by elevating blood pressure or producing nephrotoxic substances, such as reactive oxygen species and nitric oxide. These factors eventually cause glomerulosclerosis and tubular necrosis [48]. Our study results support the relationship between smoking and decreasing renal function in third-stage CKD. In Table 2, smoking with creatinine level and age with smoking account as the fourth and tenth most important risk factor, respectively.

Obesity, another risk factor of CKD, elevates blood pressure via three mechanisms: (1) activation of RAAS; (2) increasing sympathetic tone; (3) significant visceral fat compressing the kidney, and elevated blood pressure decreasing the renal function. Metabolic abnormalities like high blood sugar and abnormal lipid profile in obesity also contribute to renal impairment [48,49]. This relationship was also noted in our study. Body Mass Index with creatinine level is the seventh most important risk factor in third-stage CKD as presented in Table 3. Timely risk assessment of CKD and the increase of potential risk factors are important for preventing further kidney injury in early CKD patients.

## 5. Conclusions

The proposed auto-ML scheme auto-selects the level of each strategy to associate with a classifier, which shows an acceptable testing accuracy of 86.17%, balanced accuracy of 84.08%, the sensitivity of 90.90%, and specificity of 77.26%, precision of 88.27%, and F1 score of 89.57%. In addition, the experimental results showed that age, creatinine, hypertension, and smoking are important risk factors, which were proven in previous studies. Our automated machine learning model illustrates the possibility of assessing the combination of these risk factors under various clinical conditions. For different clinical datasets, the appropriate data preprocessing strategy, feature selection strategy, cross-validation strategy or model learning strategy can be adapted automatically. As long as the user prepares his own custom dataset with appropriate annotation. The data type of the feature can be either class or continuous, and the response should be binary. The proposed method can determine the corresponding strategy combinations and risk factors in a small number of training sessions without any manual intervention. This methodology provides essential information and longitudinal change for personalized treatment in the future.

## Figures and Tables

**Figure 1 ijerph-18-12807-f001:**
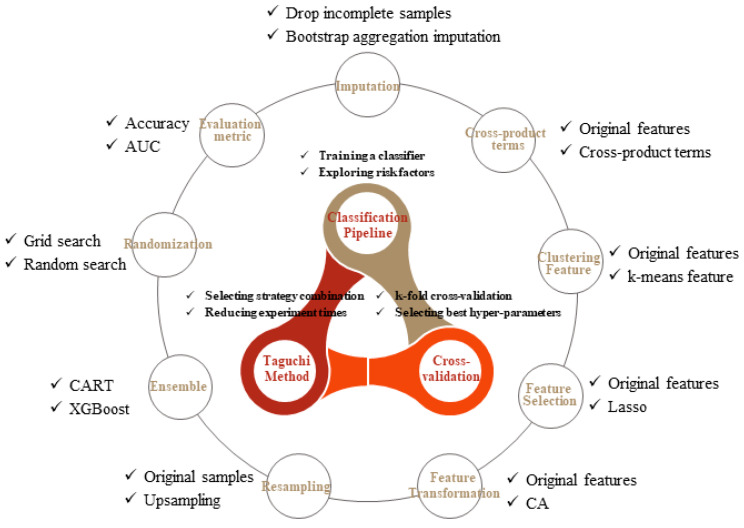
The components of proposed auto-ML scheme. CART, XGBoost, AUC, Lasso and CA are the abbreviations of classification and regression tree, eXtreme gradient boosting extreme, area under curve, least absolute shrinkage and selection operator, and correspondence analysis respectively.

**Figure 2 ijerph-18-12807-f002:**
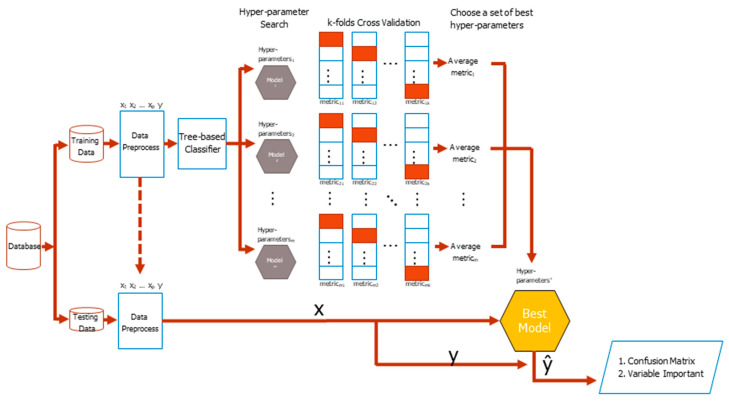
The basic flowchart of classification. ‘*’ means the select optimal set of hyper-parameters. p, m, k mean number of features, number of search times, and number of folds.

**Figure 3 ijerph-18-12807-f003:**
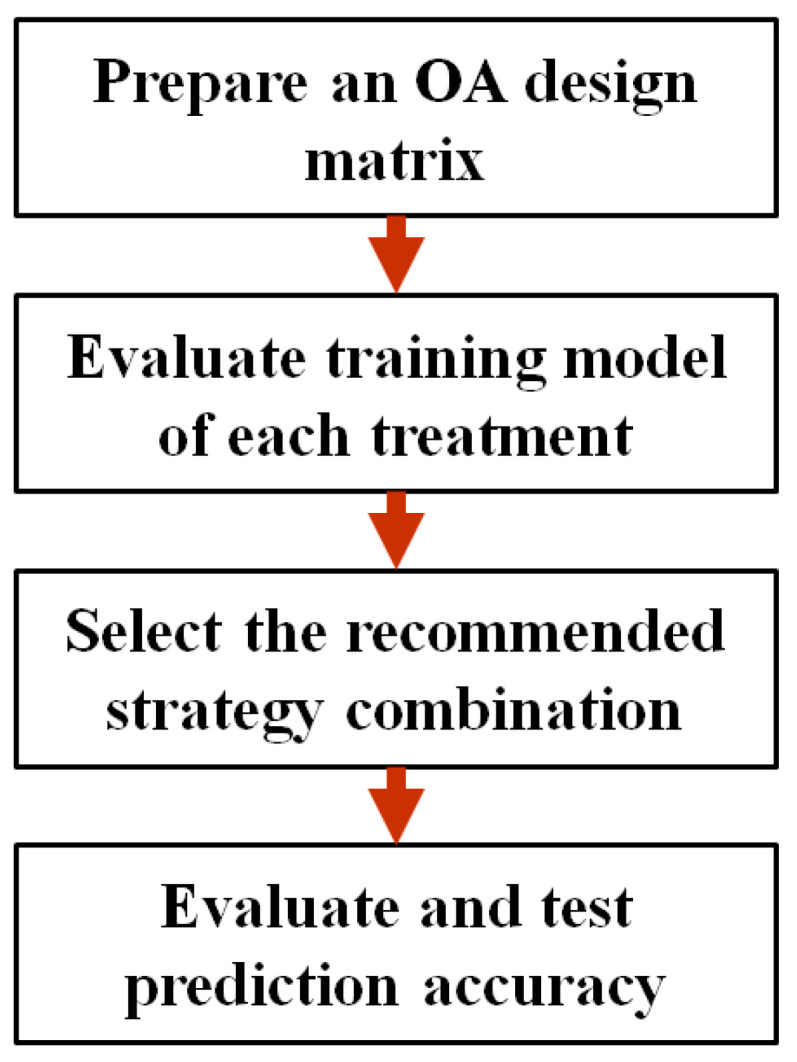
The flowchart of the proposed methodology.

**Figure 4 ijerph-18-12807-f004:**
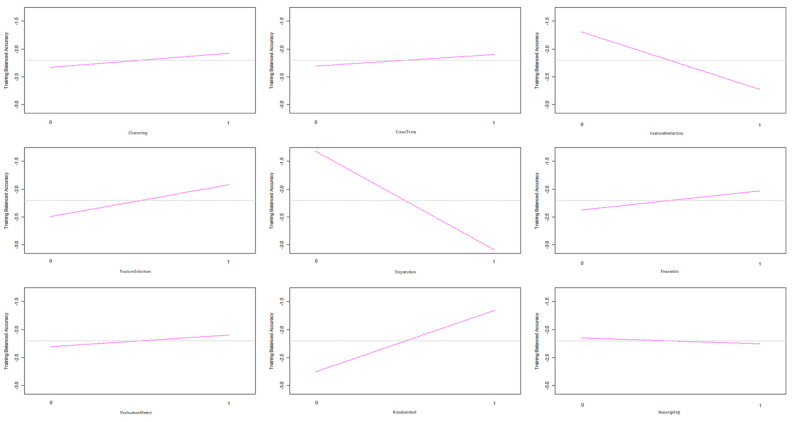
The main effect plot for each strategy.

**Figure 5 ijerph-18-12807-f005:**
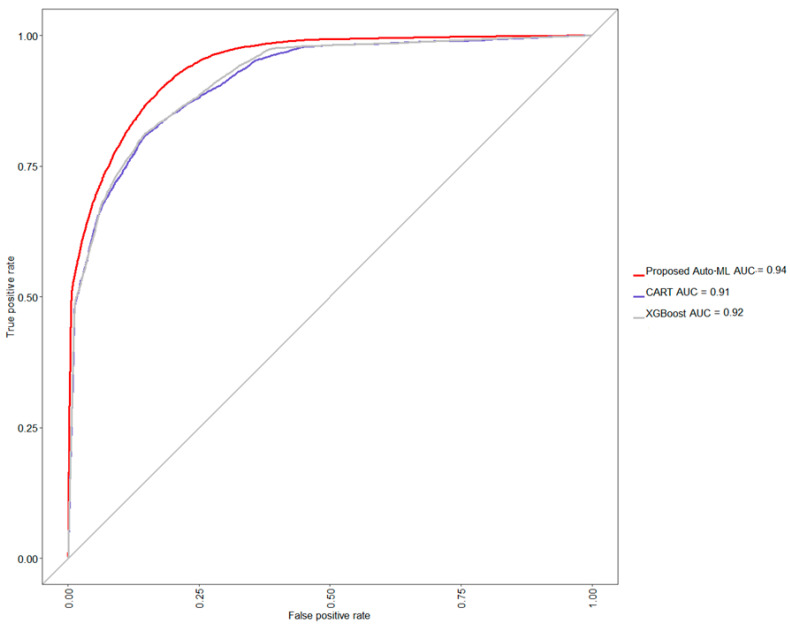
The ROC curve among different methods. Auto-ML, CART, XGBoost, and AUC are the abbreviations of automatic machine learning, classification and regression tree, eXtreme gradient boosting extreme, and area under curve respectively.

**Table 1 ijerph-18-12807-t001:** The resulting design matrix for third-stage CKD deterioration classification.

Factors (Our Strategies)									Training Balanced Accuracy
Clustering	Cross Term	Feature Reduction	Feature Selection	Imputation	Ensemble	Evaluation Metric	Randomized	Resampling	
0	0	0	0	0	0	0	0	0	0.8451
0	0	0	0	0	1	1	1	1	0.8726
0	0	1	1	1	0	0	0	1	0.5589
0	1	0	1	1	0	1	1	0	0.8538
0	1	1	0	1	1	0	1	0	0.6565
0	1	1	1	0	1	1	0	1	0.8642
1	0	1	1	0	0	1	1	0	0.8503
1	0	1	0	1	1	1	0	0	0.6546
1	0	0	1	1	1	0	1	1	0.8845
1	1	1	0	0	0	0	1	1	0.8646
1	1	0	1	0	1	0	0	0	0.8570
1	1	0	0	1	0	1	0	1	0.6531

**Table 2 ijerph-18-12807-t002:** Performance comparison of three methods in the experiment (in percentage).

Method	Accuracy	Balanced Accuracy	Sensitivity	Specificity	Precision	F1 Score
CART	84.14	82.02	88.97	75.06	87.04	87.99
XGBoost	83.82	79.39	93.86	64.92	83.44	88.34
Proposed auto-ML scheme	86.17	84.08	90.90	77.26	88.27	89.57

**Table 3 ijerph-18-12807-t003:** The top 10 ranked feature importance for CKD third stage progression.

Rank	The Combination of Risk Features	Feature Importance
1	Age × Creatinine	0.3001
2	Creatinine	0.2236
3	Hypertension × Creatinine	0.0921
4	Smoking × Creatinine	0.0723
5	Creatinine × Comorbidity	0.0542
6	Diastolic Pressure × Creatinine	0.0354
7	BMI × Creatinine	0.0337
8	Age	0.0188
9	Systolic Pressure × Creatinine	0.0181
10	Age × Smoking	0.0132

## Data Availability

Data are available from the Institutional Review Board of Chung Shan Medical University Hospital for researchers who meet the criteria for access to confidential data. Requests for the data may be sent to the Chung Shan Medical University Hospital Institutional Review Board, Taichung City, Taiwan (e-mail: irb@csh.org.tw).

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
