# Peer review of "Development of a Longitudinal Diagnosis and Prognosis in Patients with Chronic Kidney Disease: Intelligent Clinical Decision-Making Scheme"

_ijerph, 2021, doi:10.3390/ijerph182312807_

Round 1

Reviewer 1 Report

This article describes the development and testing of a machine learning model to evaluate the classifier for risk of CKD onset and prognosis using a large patient data base. The main strength of the paper is confirming that the AL methodology “provides an easy way to auto-find out a suitable model for a given dataset”. 

The results of the model, as presented and discussed, do not contribute much to the current knowledge on CDK onset or progression.  The results re-verify the association between well-established risk factors (e.g. hypertension, smoking) individually and in combination, on CDK risk.  The relative level of risk for each variable/combination of variables (as interactions) is presented for the CDK data set, but the applicability or significance of this finding, either for the population of the test data set or other data sets, is not described in the discussion.   

Author Response

The method presented in this study has the opportunity to adapt to various clinical conditions. For different clinical datasets, it is able to auto-select suitable data pre-processing strategies, feature engineering strategies, cross-validation strategies or model learning strategies as possible. As long as the users prepare their custom dataset and makes proper annotations, the proposed method can determine the corresponding strategy combination and risk factors in a small number of training times without any human intervention.

Reviewer 2 Report

It's a good idea to use machine learning to predict the possibility of evaluation for the effectiveness of one or a combination of those risk factors and provide essential information for personalized treatment in the future. In this paper, the authors found that auto-ML scheme can auto-select the level of each strategy to associate with a classifier which finally shows an acceptable testing accuracy, balanced accuracy, sensitivity and specificity. I would like to ask few questions, which would be great if it can be addressed.

  1. Creatine is not an accurate marker for the stage of CKD, it would be great if the authors can use the change of creatine for the renal injury and risk fact for progressive CKD?
  2. It’s well known that the renal function declined with age. I was wondering if the authors consider of age adjusted creation for the evaluation methods?
  3. Proteinuria is the hiding markers for the progression of CKD, it’d be great if authors can include it as well for the test mode.

Author Response

  1. Creatine is not an accurate marker for the stage of CKD, it would be great if the authors can use the change of creatine for the renal injury and risk fact for progressive CKD?

Response:

In general, CKD is divided into five stages based on estimated Glomerular Filtration Rate (eGFR). We agree that eGFR is the good standard for CKD diagnosis, but eGFR is estimated from age, gender, and creatinine. In consideration of patterns of CKD progression, it is critical to conduct risk diagnosis and prognosis for CKD patients. In this study, we would like to discuss creatinine directly except estimated value. Moreover, CKD risk factors, such as hypertension, age, UPCR, smoking, obesity and so on. This methodology provides essential information and longitudinal change for personalized treatment in the future. We have revised the Conclusion section as suggested by the reviewer.

  1. It’s well known that the renal function declined with age. I was wondering if the authors consider age-adjusted creation for the evaluation methods?

Response:

We thank the reviewer for allowing us to further explain this issue. Our machine learning is designed for providing essential information for personalized treatment in the future, and age is an important parameter in personalized treatment. Hence, we would like to discuss age alone and age with other risk factors rather than age-adjusted parameters.

  1. Proteinuria is the hiding markers for the progression of CKD, it’d be great if authors can include it as well for the test mode

Response:

We thank the reviewer for reminding us this suggestion. Proteinuria is another good standard for CKD diagnosis, but our aim is to evaluate other related risk factors expect the parameters in CKD diagnosis. Thank you for your valuable comment. It has been included as one of further research direction in Discussion section.

“Obesity, another risk factor of CKD, elevates blood pressure via three mechanisms: (1) activation of RAAS; (2) increasing sympathetic tone; (3) significant visceral fat compressing the kidney, and elevated blood pressure decreasing the renal function. Metabolic abnormalities like high blood sugar and abnormal lipid profile in obesity also contribute to renal impairment [52,53]. This relationship was also noted in our study. Body Mass Index with creatinine level is the seventh most important risk factor in third-stage CKD as presented in Table 2. Timely risk assessment of CKD and the increase of potential risk factors are important for preventing further kidney injury in early CKD patients.”

Reviewer 3 Report

The authors suggest a machine learning model for the analysis of Chronic Kidney Disease using clinical data on more than 50,000 patients. They apply Taguchi method for the experiment design to significantly decrease time complexity of their algorithm. 

The obtained forecasting model demonstrates sensitivity of 90%, specificity of 77% and balanced accuracy of 84% on the test sample, which is better than for state of the art models.

From machine learning point of view, the paper is interesting and well-written. I have only some minor comments.

Page 6, formula (1): quantity $y$ is not specified.

Page 9, Figure 5: this figure looks like ROC curve, not PR curve.

The authors should add information on the estimated values of precision and F1 measures.

Author Response

Page 6, formula (1): quantity $y$ is not specified.

Response:

The quantity y in formula (1) is specified as “training balanced accuracy” in Section 2.1. The modified formula is shown in below.

(1)

Page 9, Figure 5: this figure looks like ROC curve, not PR curve.

Response:

Thanks for suggestion to improve our work. The term “PR curve” is modified as “ROC curve”. It has been rewritten as “A graphical comparison via a receiver operating characteristic (ROC) curve is also shown in Figure 5, confirming that the proposed method provides an easy way to auto-find out a suitable model for a given dataset.”

The authors should add information on the estimated values of precision and F1 measures.

Response:

The information of the estimated values of precision and F1 measures has been added in Abstract, Discussion and Conclusion sections.

The relevant description is shown in below.

Abstract: Previous studies on CKD patients have mostly been retrospective, cross-sectional studies. Few studies have assessed the longitudinal assessment of patients over an extended period. In consideration of the heterogeneity of CKD progression. It's critical to develop a longitudinal diagnosis and prognosis for CKD patients. We proposed an auto Machine Learning (ML) scheme in this study. It consists of four main parts: classification pipeline, cross-validation (CV), Taguchi method and improve strategies. This study includes datasets from 50,174 patients, data were collected from 32 chain clinics and three special physical examination centers, between 2015 and 2019. The proposed auto-ML scheme can auto-select the level of each strategy to associate with a classifier which finally shows an acceptable testing accuracy of 86.17%, balanced accuracy of 84.08%, sensitivity of 90.90% specificity of 77.26%, precision of 88.27%, and F1 score of 89.57%. In addition, the experimental results showed that age, creatinine, high blood pressure, smoking are important risk factors, and has been proven in previous studies. Our auto-ML scheme light on the possibility of evaluation for the effectiveness of one or a combination of those risk factors. This methodology may provide essential information and longitudinal change for personalized treatment in the future.

  1. Discussion

Based on the optimal model selected throughout the training process described above, approximately 20,068 cases were further fed for testing the model’s performance of the proposed method. The rate of third-stage CKD progression in testing data is also approximately 34.69%. The proposed auto-ML scheme auto-selects the level of each strategy to associate with a classifier, which finally shows an acceptable testing accuracy of 86.17%, balanced accuracy of 84.08%, a sensitivity of 90.90%, specificity of 77.26%, precision of 88.27%, and F1 score of 89.57%. Comparing the performance of two naive situations, i.e., only CART or XGBoost classifier is used and none strategy is adopted, the CART yields a lower testing accuracy of 84.14%, balanced accuracy of 82.02%, sensitivity of 88.97%, specificity of 75.06%, precision of 87.04%, and F1 score of 87.99%, whereas the XGBoost also yields a lower testing accuracy of 83.82%, balanced accuracy of 79.39%, sensitivity of 93.86%, specificity of 64.92%, precision of 83.44%, and F1 score of 88.34%. A graphical comparison via a receiver operating characteristic (ROC) curve is also shown in Figure 5, confirming that the proposed method provides an easy way to auto-find out a suitable model for a given dataset.

  1. Conclusions

The proposed auto-ML scheme auto-selects the level of each strategy to associate with a classifier, which shows an acceptable testing accuracy of 86.17%, balanced accuracy of 84.08%, the sensitivity of 90.90%, specificity of 77.26%, precision of 88.27%, and F1 score of 89.57%. In addition, the experimental results showed that age, creatinine, hypertension, and smoking are important risk factors, which were proven in previous studies. Our model of auto-ML sheds light on the possibility of evaluation for the effectiveness of one or a combination of those risk factors. This methodology provides essential information and longitudinal change for personalized treatment in the future.

Round 2

Reviewer 1 Report

The authors reply is insufficient to address

the concerns in from initial review.

At a minimum, the authors need substantial

revision of their discussion to adequately compare and contraer

 their model in context of other models

and methods to kidney disease and progression.

Author Response

Thanks for your suggestions. To further illustrate the comparison of the methods, we add the following descriptions and table 2 in the revised version.

“In Table 2, the proposed auto-ML scheme auto-selects the level of each strategy to associate with a classifier, which finally shows an acceptable testing accuracy of 86.17%, balanced accuracy of 84.08%, a sensitivity of 90.90%, and specificity of 77.26%, precision of 88.27%, and F1 score of 89.57%. Further, comparing the performance of two naive situations, i.e., only CART or XGBoost classifier is used and none strategy is adopted, the CART yields a lower testing accuracy of 84.14%, balanced accuracy of 82.02%, sensitivity of 88.97%, and specificity of 75.06%, precision of 87.04%, and F1 score of 87.99%, whereas the XGBoost also yields a lower testing accuracy of 83.82%, balanced accuracy of 79.39%, sensitivity of 93.86%, and specificity of 64.92%, precision of 83.44%, and F1 score of 88.34%. From this model comparison experiment, it can be seen that the classification accuracy of CART has reached the level of about 84%. Compared with CART, the classification accuracy of XGBoost has decreased, and the level of specificity has also been sacrificed. In this study, XGBoost was selected as the basic classifier, and with the help of other strategies, it can further improve the classification accuracy rate by about 2% and the predictions will not be biased towards the majority class.”

Table 2. Performance comparison of three methods in the experiment (in percentage).

Method

Accuracy

Balanced accuracy

Sensitivity

Specificity

Precision

F1 score

CART

84.14

82.02

88.97

75.06

87.04

87.99

XGBoost

83.82

79.39

93.86

64.92

83.44

88.34

Proposed auto-ML scheme

86.17

84.08

90.90

77.26

88.27

89.57

Round 3

Reviewer 1 Report

The authors' response presents a strength of the model in that is "adaptable to various clinical conditions" to identify combinations of strategies and risk factors.  A brief description of the type of preparation and annotation the user must do to adapt the model would be appropriate and make the article more accessible to a broader range of practitioners and researchers.  

Author Response

Thanks for your suggestion. To further illustrate the type of preparation and annotation, we add the following descriptions in the revised version.

“Our automated machine learning model illustrates the possibility of assessing the combination of these risk factors under various clinical conditions. For different clinical datasets, the appropriate data preprocessing strategy, feature selection strategy, cross-validation strategy or model learning strategy can be adapted automatically. As long as the user prepares his own custom dataset with appropriate annotation. The data type of the feature can be either class or continuous, and the response should be binary. The proposed method can determine the corresponding strategy combinations and risk factors in a small number of training sessions without any manual intervention.”

Please refer to the Conclusions Section of the revised manuscript.